# Promising Strategies for the Management of Burn-Wound-Associated Pruritus

**DOI:** 10.3390/ebj6010002

**Published:** 2025-01-24

**Authors:** Mayer Tenenhaus, Hans-Oliver Rennekampff

**Affiliations:** 1Independent Researcher, San Diego, CA 92107, USA; 2Plastic Surgery, Rhein-Maas Klinikum, 52146 Würselen, Germany

**Keywords:** post burn pruritus, itching, treating pruritus, burn wound, burn scar

## Abstract

Patients who have been injured by burns often suffer from persistent and debilitating post burn pruritus. Despite a myriad of therapeutic interventions and medications, this complex condition remains particularly difficult to ameliorate. Recently, a new generation of antipruritic medications has demonstrated clinical success in managing pruritus in a number of dermatologic, nephritic and hepatic disease states, targeting unique aspects of the pruritic pathways. While specific trials demonstrating efficacy and safety are currently lacking, the purported mechanisms of action and similarities to the targeted inflammatory markers, pruritogens and neural pathways of these new medications, in concert with clinical evidence, hold promise for burn patients.

## 1. Introduction

Tragically, many patients injured by burns suffer terribly from post burn pruritus. While our understanding of the complex physiologic and pathophysiologic interactions and expressions associated with wound healing have added much to the quality of care afforded to those in our care, effective therapeutic strategies for ameliorating post burn pruritus continue to be a particular challenge. Clinical research relating numerous inflammatory markers, cytokines, cascades and agents expressed both locally at burn wound sites and scars, as well as an evolving understanding of the fundamental systemic and neurologic pathways, have afforded new insights and approaches that can address post burn pruritus.

Several of the pruritogens and markers identified and expressed in a number of challenging dermatologic diseases such as atopic dermatitis [1,2], prurigo nodularis (PN) [3] and notalgia paresthetica [4] have been identified, and there are overlaps in several cases with those expressed in burn wounds and burn scars [5]. This suggests potential or shared pathophysiologic links and targets. Recently, a new generation of approved medications has shown clinical benefit for pruritus in the aforementioned dermatologic, hepatic, renal and autoimmune disease states. It is reasonable to consider these rapidly evolving strategies to hold promise for sufferers of post burn pruritus.

The incidence of post burn pruritus is difficult to definitively establish; however, there is little doubt that its incidence is concerning, with journal articles relating rates of as high as 80 to 100% [5]. Post burn pruritus can present within days of the insult and can persist for years [6].

Efforts to establish predictors for post burn itching have generally identified individuals in younger age groups, women and individuals with dry skin, as well as patients with raised or thickened scars [6]. Considerations as to which specific types and distributions of burn injuries are more prone to the development of post burn pruritus remain illusive, as on review, the descriptions often vary and contradict each other. Several reviews relate that burns to the lower extremities, as well as injuries which ultimately manifest as raised and thickened scars, are most likely to be associated with post burn pruritus [5]. Facial burn injury distributions are considered unlikely to result in significant post burn pruritus in some reviews, while other publications specifically contradict this finding [5,7]. Regardless, post burn pruritus commonly manifests in a significant percentage of individuals who have suffered a burn injury, and this complex and often debilitating complication can prove extremely challenging to manage or treat. The negative impact of post burn pruritus on the sufferer’s psychosocial well being, work and sleep has been well described [8]. The inevitable physiologic response to this unabating pruritus is scratching, which promotes further inflammation, delayed wound healing and a propensity to the development of pathologic scarring.

Despite the litany of available treatment strategies and agents (Table 1), their overall clinical efficacy remains lacking. A recent Cochrane review of pertinent publications regarding 21 different interventions employed for the treatment of post burn pruritus found at best moderate- to low-certainty evidence for their efficacy [9]. Thick scar formation, such as that seen with PN and pathological burn scars, can lessen the efficacy and penetration of topically applied modalities [10]. This disappointing reality parallels that observed clinically by both senior authors of this text.

### 1.1. What Do We Know About Pruritus and Specifically Post Burn Pruritus?

Pruritus is a rather complex entity that has not yet been fully elucidated. It is understood that pruritus results from multiple pathophysiologic pathways, predeterminants and intrinsic responses to a variety of expressed products and stimulants. As such, it is not surprising that attempts to treat this entity have proven so challenging. The best-described pathways and stimuli response patterns associated with the development of pruritus include local and systemic responses to pruritogens and cytokines, as well as autonomic, sensory and cognitive neurobiological pathways.

Classically, pruritus was thought to either be a peripheral or centrally mediated entity. Direct, translational and transcriptional interactions likely blur this distinction. Peripheral pruritus has generally been considered to be an inflammatory process whereby IgE and complements stimulate the mast cells in the skin dermis and the subcutaneous adipose tissue to release proteases and histamine, which then bind to itch receptors in the dermo-epidermal junction. It is for this reason that medications considered to aid in peripheral pruritus traditionally included topical steroids, antiallergic medications and antihistamines. Activated itch signals transmitted via C fibers to the spinal cord are then recognized as pruritus in the brain [11,12].

Central pruritus is thought to result from an imbalance in the activation or expression of the μ- and κ-opioid receptors [13]. μ-opioid receptor agonists trigger central pruritus, while κ-opioid receptor agonists suppress central pruritus. Under normal unstimulated conditions, μ- and κ-opioid receptor activities maintain equilibrium. Injured keratinocytes and nerve cells produce β-endorphin and dynorphin, endogenous opioids. β-endorphin, the ligand of the μ-opioid receptor, induces pruritus, whereas dynorphin, the ligand of the κ-opioid receptor, inhibits pruritus. This neuroimmune interface is similarly modulated and likely driven by stress. Recognizing this balanced integral relationship, it becomes easier to appreciate that strategies that target peripheral pruritus might prove less effective in central pruritus. Ultimately, central and peripheral nervous system responses trigger scratching [14].

While a comprehensive understanding of these complex pathways and interactions far exceeds that which is penned in this text, we can, however, focus on primary cytokines which are known to be involved in the development and expression of post burn pruritus, as well as neuromodulators and neural-signaling-related receptors.

The primary cytokine associated with burn pruritus is generally considered to be Interleukin-31 (IL-31), often referred to as a key “pruritogen” directly responsible for triggering itch sensations in the skin, burn wounds and wound healing processes. IL-1β, IL-6 and TNF-α may also contribute to the inflammatory response leading to pruritus in burn injuries [5].

### 1.2. Newer-Generation Medications That Target Responsible Cytokines

Nemolizumab is a subcutaneously administered humanized monoclonal antibody against interleukin-31 receptor A, which is involved in pruritus and inflammation in atopic dermatitis [2]. In a phase 3 trial involving patients with atopic dermatitis, nemolizumab used along with topical agents resulted in a significant reduction in pruritus as compared to a placebo plus topical agents, with no increase in adverse reactions when compared to the placebo at 16 weeks. In a separate study, nemolizumab did not show a significant benefit over the placebo for patients suffering from uremic pruritus.

Vixarelimab is a new human monoclonal oncostatin M receptor Beta antibody. In a 2023 phase 2a randomized double-blind placebo-controlled trial for patients suffering from moderate to severe itching due to prurigo nodularis (PN), Vixarelimab demonstrated a rapid reduction in pruritus, with almost clear skin in one-third of the patients by week 8. The drug-related treatment-emergent adverse event rates were noted to be similar between the placebo and treatment groups [15].

Dupilumab is a human monoclonal antibody that blocks the shared receptor component for interleukin (IL)-4 and IL-13. In a phase 3 double-blind and randomized controlled clinical trial of patients with prurigo nodularis published in 2023, the patients were randomized to receive dupilumab or a placebo subcutaneously every 2 weeks for 24 weeks [16]. The authors concluded that dupilumab showed benefits in the Worst Itch Numeric Rating Scale (WI-NRS), the Skin Pain NRS and the Dermatology Life Quality Index (DLQI), which increased progressively over 24 weeks of treatment. Dupilumab demonstrated clinically meaningful and statistically significant improvements in itching and skin lesions versus those with the placebo in PN, and no blood work testing was required prior to its administration. Its safety was consistent with dupilumab’s known safety profile [16]. IL-4 and IL-13 are considered key moderators in the development of pathologic scars such as those seen post burn, which might suggest the potential role of this drug [17]. Interestingly, a recent paper by Mulder et al. related that while TNF-α, IFN-γ, IL-1β, IL-6, IL-8 and Monocyte chemoattractant protein-1 were all increased in burn tissues, typical M2 macrophage factors like IL-4, IL-10 and IL-13 were not detected [18]. This somewhat conflicting data must, however, be tempered by the fact that agents like dupilumab clinically demonstrate broad efficacy against a number of conditions with variable elevations in IL-4 and/or IL-13, such as prurigo nodularis and chronic pruritus of unknown origin. It has been suggested that blocking even the physiological levels of IL-13 and IL-4 may be effective in alleviating neural itching because these expressed cytokines promote neural hypersensitivity to multiple pruritogens [4,19,20]. Future studies are needed to understand this relationship better and its potential use in modulating post burn pruritus.

### 1.3. Newer-Generation Medications That May Aid in Targeting the Neurologic Aspects of Pruritus

Non-histaminergic pathways have traditionally been considered the predominant pathophysiologic pathway in chronic itch conditions. As such, it is easier to understand why many traditional histamine-directed pruritus therapies have failed [21]. Cytokines and their associated pathways play a significant role in the interrelationship between the immune and sensory systems. Key cytokines, particularly those relevant to pruritus, have been shown to signal through a family of Janus kinase/signal transducer and activator of transcription (JAK/STAT) molecules regulating gene expression. This intrinsic relationship and their role in both pain and itching promote potential targeting strategies for a new generation of medications [21,22,23].

Ruxolitinib is a selective inhibitor of Janus kinase 1 (JAK1) and Janus kinase 2 (JAK2). It is available as both an oral tablet and a cream. Ruxolitinib is used to treat steroid-resistant acute graft-versus-host disease, as well as polycythemia vera and myelofibrosis. In atopic dermatitis, it is thought to act by suppressing cytokine signaling. A phase 2 study published in 2020 compared the application of a topical cream two times a day (BID) vs. a vehicle, concluding that Ruxolitinib demonstrated an improvement in the Eczema Area and Severity Index, as well as a reduction in pruritus, without clinically significant application site reactions [24].

Κ-opioid receptors bind opioid-like compounds in the brain and have been shown to mediate mood, pain and consciousness. Κ-opioid receptor agonists have been shown to aid in suppressing itching. Difelikefalin is a peripherally restricted κ-opioid receptor agonist, and as such, it lacks the concerning CNS side effects seen in centrally active κ-opioid receptor agonists. It is currently indicated for moderate to severe pruritus associated with chronic kidney disease and in adult hemodialysis patients. In Japan, the drug is approved for patients on peritoneal dialysis struggling with pruritus. In a 2022 review of the currently available clinical trials, a safety and efficacy assessment of difelikefalin in patients with uremic pruritus was undertaken. In their review, the authors concluded that Nalfurafine hydrochloride improved quality of life per the Skindex score, as well as causing an improvement in the 5-D itch scale, suggesting that difelikefalin was effective in reducing pruritus in the patients studied [25].

Nalfurafine is a selective κ-opioid receptor agonist used clinically as an antipruritic medication and approved for use in Japan for uremic pruritus. It is administered as an intravenous infusion during dialysis. Nalfurafine hydrochloride has also been approved for pruritus in patients with chronic liver disease in whom the pruritus is generalized, as opposed to localized. This form of pruritus is extremely refractive to most treatments. Its efficacy was confirmed in a 2022 randomized, double-blinded trial by Kawano et al. in which the authors found that Nalfurafine hydrochloride not only shortened the pruritic period but also seemed to lessen the development of advanced fibrosis [26,27,28,29]. There is now an oral formulation of this medication; however, at the time of submission, pertinent data are lacking.

Notalgia paresthetica is a localized and chronic condition characterized by intense burning and itching affecting the nerves of the torso, commonly in the inter-scapular region. A phase 2, double-blind, placebo-controlled trial published in 2023 randomly assigned NP patients with moderate to severe pruritus to receive BID oral difelikefalin vs. a placebo for 8 weeks. The authors found that difelikefalin resulted in modestly greater reductions in their itch intensity scores than the placebo; however, their results are tempered by several associated adverse events, notably headaches, dizziness, constipation and increased urine output [30].

## 2. Discussion

Despite a myriad of therapeutic strategies, medications and interventions, chronic and often unrelenting pruritus continues to severely and negatively impact quality of life for a great many patients who have suffered burn injuries. Appreciation of various pathophysiologic mechanisms and interrelated pathways associated with the development and manifestation of pruritus has led to new targeted approaches.

An evolving understanding of the complex, multifaceted and interrelated facets of the itch–scratch cycle both complicates and yet opens up new potential targeting strategies for intervention.

Experiences gained from randomized and controlled clinical trials specifically addressing pruritus with a variety of different clinical presentations (dermatologic, renal and hepatic) may well hold promise for patients suffering from post burn pruritus. An internet and PubMed search of clinical trials employing the following key words—burn pruritus, post burn pruritus, itching burn wound, clinical trials burn pruritus—identified a single clinical trial, NCT06226610, entitled “Dupixent in Adults with refractory post burn pruritus in an ambulatory clinic”. This study is currently in the recruiting phase.

The ultimate choice of antipruritic therapy, as with all clinical interventional strategies, requires an individualized approach founded on reliable and validated data, experience, a risk and benefit analysis, cost, insurance coverage and availability.

## 3. Conclusions

Post burn pruritus is a common and serious complication of burn injuries which severely lowers the quality of life for our patients. While there is some solace to be gained from the knowledge that post burn pruritus tends to ease over time, the reality is that many suffer with itching long after their burn injury. Compromised sleep and difficulties in concentrating negatively impact their participation in social, intellectual and work activities. These untoward effects manifest as both direct and indirect consequences of the pathophysiologic state, medications and interventional measures. Patients relate associated anxiety and depression, further contributing to the injury response state.

Strategies that target the histaminergic pathways and the use of nonspecific anti-inflammatory agents, as well as traditional non-pharmacological methods, afford some solace; however, there remains little doubt that the pathophysiology of post burn pruritus is more complex that presently understood.

There is no single standard treatment that is consistently effective at this time. It has been suggested that combination approaches targeting peripheral as well as central agents will likely prove most efficacious [31]. The new generation of medications introduced above has shown promise in modulating similar pathophysiologic processes by virtue of newly appreciated pathways and targets. As such, in the view of the authors, these strategies and future physiologic derivatives might hold promise for patients suffering from post burn pruritus. Without the availability of well designed and controlled trials, it would, at this time, be premature to advocate beginning with these new-generation antipruritic medications. That said, time will tell.

## Figures and Tables

**Table 1 ebj-06-00002-t001:** Commonly described therapeutic interventions and strategies for the management of post burn pruritus.

Topical Agents	Oral Medications	Neurologic Pathway And Analgesic Agents	Non-pharmacological Interventions	Common Home Remedies
CORTICOSTEROIDS	STEROIDS	GABAPENTIN	SOMATOSENSORY FEEDBACK REHABILITATION	BATHS, i.e., comfortable- or cool-temperature baths
COOLANTS, i.e., menethol, camphor, icilin	ANTIHISTIMINES, i.e., diphenhydramine, cetirizine, loratadine or hydroxyzine	PREGABALIN	PHYSICAL TREATMENTS, i.e., extracorporeal shock wave therapy (ESWT), massage	BATH ADDITIVES, i.e., oatmeal, powdered milk or a variety of oils and moisturizers, such as baby oil
DOXEPIN CREAM	ASCORBIC ACID	ANTIDEPRESSANTS, i.e., tricyclic antidepressants, selective serotonin reuptake inhibitors, serotonin, norepinepherine reuptake inhibitors, doxepin	ELECTRO-CONVULSIVE THERAPY	COLD, i.e., topically applied cool compresses or having cool air directed at the offending site using a fan or an air conditioner
TOPICAL LOCAL ANESTHETICS, i.e., Lidocaine, TKAL (compounded 10% Ketamine–5% Amitriptyline–5% Lidocaine as a LIPODERM CREAM)		BOTULINUM TOXIN	TENS (transcutaneous electrical nerve stimulation)	DISTRACTION TECHNIQUES, i.e., participating in hobbies, activities, stretching and exercise
		ANTI-CYTOKINE AGENTS	SILICONE GEL SHEETING	
		NEUROKININ-1 INHIBITORS	ELECTROPUNCTURE	
		PAR-2 INHIBITORS	NERVE RELEASE	
			LASER i.e., non-ablative, Fraxel and pulsed dye laser therapies

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
