# Peer review of "Promising Strategies for the Management of Burn-Wound-Associated Pruritus"

_2673-1991, 2025, doi:10.3390/ebj6010002_

Round 1

Reviewer 1 Report

Comments and Suggestions for Authors

Thank you for giving me the opportunity to review this perspective paper. The topic of this review is certainly one of interest to the burns community, and one where further evidence is required. The authors have chosen to review the current understanding of the pathophysiology of pruritus in burns and the state of treatment options currently available and those that are emerging in other fields. Their summary of the mechanisms is of high-quality, and would be useful reading for any clinician who regularly manages burn patients.

The novel aspect of their paper is within the summary of emerging treatments for pruritus, including Nemolizumab, Vixarelimab, Dupilumab, Ruxolitinib and Nalfurafine, that have seen success in other conditions such as atopic dermatitis, prurigo nodularis and notalgia paresthetica. The overlap of identified pruritogens between these conditions and those found in post-burn pruritus is of interest and should definitely spark interest in possible trials of these treatments for this patient cohort.

On that note, the authors have not commented on whether there are any planned or registered trials of any of these medications for burns pruritus. On a cursory search of https://clinicaltrials.gov/ there is at least one phase 2 RCT of Dupilumab (Dupixent) for Post-Burn Pruritus (NCT06226610) that is soon to start recruiting. There may be others, and this information would be important for researchers reading this paper to know, especially given that one of the intended outcomes of this manuscript appears to be a ‘call to arms’ to improve the evidence base for post-burn pruritus.

There are some issues that the authors should address in the current form of their manuscript. Firstly, the authors use acronyms that are not initially defined. These include ‘TENS’ (Transcutaneous electrical nerve stimulation) on page 2, line 92, and BID on page 5, line 208. Moreover, in the ‘currently employed strategies’ on page 2, the authors have a subheading ‘Biologics’ and then go on to describe Gabapentin and Pregabalin. Neither of these drugs are considered ‘biologics’ in the traditional sense, and I wonder if the authors could describe further what they mean by this and include the examples of biologics. I wonder whether they intended for these drugs to go in the ‘Neurologic pathway and analgesic agents’ section? The authors also include ‘histamine antagonists’ in this section, but have already included ‘antihistamines’ in ‘oral medication’. This appears to be repetition of the same class of drug. I also wonder whether it would be more appropriate to put topical silicone in the ‘non-pharmalogic treatments’ section, rather than ‘topical agents’ section, given that its mechanism of action is to prevent moisture loss by reducing the permeability of healing burnt skin. The authors have broadly included ‘antidepressants’, but it would be more appropriate to included specifics. I wonder whether this whole summary of currently employed strategies may be better formatted as a table, with examples and actions, thereby giving readers a valuable resource to go back to when they need an overview of this topic?

Overall, I think that this perspective paper is one of interest to the readership of EBJ and would be useful reading for the burns community. The emerging treatments in other fields and their possible overlap with burns pruritus is exciting and will hopefully stimulate high-quality research to improve the experience of patients who have suffered burns.

Reviewer 2 Report

Comments and Suggestions for Authors

Thank you for the opportunity to review the article „ Promising strategies for the management of burn wound associated pruritus". The topic is interesting and important.
The style of the text is very flourishing, like that of scientific text. For example, “Tragically many patients injured by burns suffer terribly from post-burn pruritus.” I found it confusing. Discussion is very short.
In my opinion, the article does not meet the criteria of a scientific article. In many fragments, the sources of cited research are missing (for example lines 47-55; 108-115; 150-151).
I do not know what the aim of the manuscript is. In the literature, there are many good-quality articles presenting the topic in a more structured and thematic way. I encourage the Authors to rewrite the text, filling in the missing sources thoroughly.
In my opinion, unfortunately, the manuscript in its current form should be rejected.

Comments on the Quality of English Language

The style of the text is very flourishing, like that of scientific text. For example, “Tragically many patients injured by burns suffer terribly from post-burn pruritus.” I found it confusing.

Reviewer 3 Report

Comments and Suggestions for Authors

Review provided in attached document.

Round 2

Reviewer 1 Report

Comments and Suggestions for Authors

Thank you for giving me the opportunity to review the revision of the interesting paper looking at the treatment of post-burn pruritus and possible future treatment options. 

The authors have addressed all points raised in the previous review, and those made by the other reviewers. The quality and presentation of the content has improved with these changes, and I believe that this paper would be of interest to the readership of EBJ.